# Plant Peroxisome-Targeting Effector MoPtep1 Is Required for the Virulence of *Magnaporthe oryzae*

**DOI:** 10.3390/ijms23052515

**Published:** 2022-02-24

**Authors:** Na Ning, Xin Xie, Haiyue Yu, Jie Mei, Qianqian Li, Shimin Zuo, Hanxiang Wu, Wende Liu, Zhiqiang Li

**Affiliations:** 1Key Laboratory of Agricultural Microbiology, College of Agriculture, Guizhou University, Guiyang 550025, China; ningna960328@163.com (N.N.); xiexin2097757@163.com (X.X.); 2State Key Laboratory for Biology of Plant Diseases and Insect Pests, Institute of Plant Protection, Chinese Academy of Agricultural Sciences, Beijing 100193, China; yuhaiyueo@126.com (H.Y.); qianyu92@foxmail.com (J.M.); wuhanxiang@caas.cn (H.W.); 3College of Life Sciences, Nanjing Normal University, Nanjing 210023, China; liqianqian7234@163.com; 4Key Laboratory of Plant Functional Genomics of the Ministry of Education, College of Agriculture, Yangzhou University, Yangzhou 225009, China; smzuo@yzu.edu.cn

**Keywords:** *Magnaporthe oryzae*, effector protein, virulence, peroxisomes, cell death

## Abstract

Rice blast caused by *Magnaporthe oryzae* is one of the most serious fungous diseases in rice. In the past decades, studies have reported that numerous *M. oryzae* effectors were secreted into plant cells to facilitate inoculation. Effectors target host proteins to assist the virulence of pathogens via the localization of specific organelles, such as the nucleus, endoplasmic reticulum, chloroplast, etc. However, studies on the pathogenesis of peroxisome-targeting effectors are still limited. In our previous study, we analyzed the subcellular localization of candidate effectors from *M. oryzae* using the agrobacterium-mediated transient expression system in tobacco and found that MoPtep1 (peroxisomes-targeted effector protein 1) localized in plant peroxisomes. Here, we proved that *MoPtep1* was induced in the early stage of the *M. oryzae* infection and positively regulated the pathogenicity, while it did not affect the vegetative growth of mycelia. Subcellular localization results showed that MoPtep1 was localized in the plant peroxisomes with a signal peptide and a cupredoxin domain. Sequence analysis indicated that the homologous protein of MoPtep1 in plant-pathogenic fungi was evolutionarily conserved. Furthermore, MoPtep1 could suppress INF1-induced cell death in tobacco, and the targeting host proteins were identified using the Y2H system. Our results suggested that *MoPtep1* is an important pathogenic effector in rice blast.

## 1. Introduction

The plant lives in an environment threatened by a variety of pathogens in nature; thus, the plant has evolved to have a complete defense system to recognize pathogens and stimulate the immune response for pathogenic infection [1]. Plant cell-surface pattern recognition receptors (PRRs) recognize pathogen-associated molecular patterns (PAMPs) or microbe-associated molecular patterns (MAMPs) to stimulate pattern-triggered immunity (PTI), whereas pathogens secret effectors into plant cells to regulate PTI immunity. To defend against effector-triggered infection, plants have evolved resistant proteins (R proteins) to specifically recognize effectors secreted by pathogens and then trigger the ETI (effector-triggered immunity) to inhibit pathogen infection [2]. Pathogens secrete more effectors to induce plant susceptibility and facilitate pathogen infection. Meanwhile, plants evolve corresponding R proteins to recognize effectors and enhance disease resistance. As a result, effectors secreted by pathogens play very important roles in the interaction between host plants and pathogens [3]. Recent studies showed that *Pseudomonas syringae* effector HopZ1a localized in the plasma membrane and was recognized by SZE1 and SZE2 to regulate the autoimmune response of the ZAR1–ZED1 complex [4]. The calcium concentration that fluctuates in plant cells plays an important role in the earliest resistances to abiotic and biotic stresses. *Phytophthora* RXLR effector Avrblb2 localized in the plasma membrane of the plant cells that interacts with calmodulin to regulate resistance-associated Ca^2+^ signaling in plants [5]. Downy mildew caused by the *Plasmopara viticola* is one of the most serious diseases in the production of grapevine. PvAvh74, a *P. viticola* RxLR effector, localized in the nucleus can trigger the plant immune response associated with programmed cell death [6]. Two nuclear-localized effectors, MoHTR1 and MoHTR2, from the *M. oryzae* regulate host plant immunity via transcriptional reprogramming [7]. Moreover, the MoCDIP4 effector from *M. oryzae* targeted the OsDjA9–OsDRP1E protein complex in the mitochondria to regulate rice immune response [8]. In conclusion, pathogens have evolved different types of effectors targeting diverse organelles to regulate the host’s innate immunity by regulating the biological functions of each organelle in the arms race between pathogens and host plants.

The peroxisome is an important eukaryotic organelle that plays an important role in cellular metabolic processes and the balance of signal molecules, as well as being the candidate target of pathogenic effectors [9,10]. Peroxisomes have been proven to have complex biological functions by transcriptomic and proteomic studies [11]. However, its involvement in the regulation of plant immune response has not been further studied. It was found that both peroxisomes and mitochondria of human cells contain RIG-1-like receptor (RLR) receptor proteins called MAVS, as well as that there is a chronological order of the activity of MAVS proteins in these two organelles in the antiviral immune response. With the onset of the virus infection, MAVS proteins in peroxisomes induce the expression of fast non-interferon dependent defense factor ISG, thus stimulating the early immune response, and then MAVS proteins in the mitochondria activate interferon-dependent signaling pathways to conduct sustained antiviral immune response. These results indicate that peroxisome is an important target organelle for antiviral signal transduction [10]. Li reported that the mutant of JA synthase-related protein, which targets peroxisomes, resulted in the resistance reduction of tomato against insects [12]. The H_2_O_2_ produced in peroxisomes has been proven to play important roles in immune resistance to the plant-parasitic nematode *Heterodera glycines* [13]. DspA/E, an effector from plant disease *Erwinia amylovora*, was found to localize in the plant peroxisomes to regulate the immune response of plants [14]. Robin et al. found three *C. higginsianum* effectors that localized in the peroxisomes of tobacco cells, and two of them with a tripeptide signal sequence are required for typical PTS1 (peroxisomal targeting signal 1) pathway functioning in peroxisomes, which preliminarily proved that plant peroxisomes are involved in the regulation of fungal effector protein-mediated immune response, but the specific regulatory mechanism remains to be further clarified [15]. At present, the studies on plant peroxisomes are mainly concentrated on plant growth and development regulated by peroxisome-localized enzymes [16]. However, there is still little known about how the peroxisomes participate in the regulation of plant immune response; in particular, their role as organelles targeted by pathogen effectors has not been systematically researched.

Based on our previous studies, we speculated that *M. oryzae* may secrete effectors into rice cells that localize in peroxisomes, which are involved in regulating rice immune response. Here, we analyzed the subcellular localization of 20 candidate effectors in *M. oryzae* using the agrobacterium-mediated transient expression system in tobacco. MoPtep1 was found to localize in plant peroxisomes, and the expression of *MoPtep1* was significantly induced in the early stage of *M. oryzae* infection (8–36 hpi), determined via qRT-PCR method. The coding protein of *MoPtep1* was localized in the plant peroxisomes and possessed a signal peptide and cupredoxin domain. Sequence analysis demonstrated that the homology of MoPtep1 in plant pathogenic fungi was evolutionarily conserved. Gene knockout results indicated that *MoPtep1* positively regulated the pathogenicity of *M. oryzae*, while it did not affect the vegetative growth of mycelia and could inhibit INF1-induced cell death via the agrobacterium-mediated transient expression system in tobacco leaves. Lastly, three MoPtep1-interacting rice proteins were screened by the Y2H system. Our results suggested that MoPtep1 might be an important pathogenic factor and play a role in virulence in rice–*M. oryzae* interaction.

## 2. Results

### 2.1. Amino Acid Sequence Alignment and Phylogenetic Analysis of MoPtep1 in Different Species

In our previous study, 851 secreted protein genes were predicted based on RL-SAGE, MPSS, and SBS transcriptome results [17]. Among them, a candidate effector protein MGG_03946 with a typical signal peptide may localize in plant peroxisomes, predicted by SignalP-5.0 and TargetP-2.0 (thereafter named MoPtep1, peroxisomes-targeted effector protein 1). MoPtep1 contains 198 amino acids and has a typical cupredoxin (cd00920) domain (Figure 1A). MultAlin (http://multalin.toulouse.inra.fr/multalin/multalin.html, accessed on 28 March 2000) and DNAMAN software were used for sequence comparison analysis. The results of multi-sequence alignment of MoPtep1 and its 17 homologous sequences showed 32.87% identity. MEGA7.0 software was used for phylogenetic analysis. The names of homologous species and GenBank accession numbers are as follows: MoPtep1 (MGG_03946), *Pyricularia pennisetigena* (XP 029750379.1), *Fusarium oxysporum* f. sp. melonis (EXK43175.1), *Fusarium oxysporum* f. sp. cubense (TVY80199.), *Fusarium phyllophilum* (KAF5540402.1), *Fusarium verticillioides* 7600 (XP 018760771.1), *Fusarium musae* (XP 044685243.1), *Diaporthe citri* (XP 043031751.1), *Diaporthe helianthin* (POS76255.1), *Diaporthe batatas* (XP 044644403.1), *Colletotrichum truncatum* (XP 036581195.1), *Colletotrichum karsti* (XP 038741855.1), *Colletotrichum incanum* (KZL87470.1), *Colletotrichum scovillei* (XP 035331936.1), *Colletotrichum orchidophilum* (XP 022474043.1), *Phytophthora cactorum* (KAG4221687.1), *Salix suchowensis* (KAG5220372.1), and *Terriglobus roseus* (MBE7180561.1). Homologous alignment showed that the cupredoxin domain was conserved in fungi (Figure 1B). In order to better understand the evolutionary patterns of divergence between MoPtep1 and other fungal homologous proteins, we conducted a phylogenetic analysis using MoPtep1 amino acid sequences from *M. oryzae* and the above species (Figure 1C). The phylogenetic analysis indicated that the sequence protein of MoPtep1 is conserved in pathogenic fungi.

### 2.2. Expression of the MoPtep1 Gene Was Induced during Infection

*MoPtep1* was identified from the transcriptome profiles of *M. oryzae*-infected rice plant leaves [17] to further identify the expression pattern of *MoPtep1* in different growth stages, including mycelium, spore, appressoria, and invasive hyphae (8 h, 18 h, 36 h, and 48 h). The expression of *MoPtep1* was low in the mycelium and spore stages, while the expression of *MoPtep1* was significantly induced at 36 h post inoculation (hpi) (Figure 2). These results indicated that *MoPtep1* was upregulated in the infection stages, especially at 36 hpi, suggesting that *MoPtep1* may play an important role in the interaction between *M. oryzae* and rice.

### 2.3. MoPtep1 Is a Secreted Protein in M. oryzae

To confirm the function of the predicted signal peptide, the yeast signal trap method was used to perform functional analysis on the full-length sequence (FL) of the *MoPtep1* gene and the fragment without the signal peptide sequence (NS). The results showed that all the YTK12 containing pSUC2-Avr1b, pSUC2-mg87, empty pSUC2 vector, pSUC2-MoPtep1-FL, and pSUC2-MoPtep1-NS displayed normal growth in the CMD-W medium. However, in the YPRAA screen medium containing raffinose as the only carbon source, only the pSUC2-MoPtep1-FL and pSUC2-Avr1b expressing yeast strain YTK12 were demonstrating growth well. This result indicated that the signal peptide of MoPtep1 was functional (Figure 3A). In addition, we confirmed the enzymatic activity based on the reduction of the dye 0.1% TTC to the insoluble red-colored triphenylformazan, whereas the negative control yeast strains remained colorless after 0.1% TTC incubation (Figure 3B). These results suggested that the N-terminal signal peptide sequence of MoPtep1 was a functional signal peptide sequence and assisted in the secretion of MoPtep1 protein.

### 2.4. MoPtep1 Is Important for Pathogenicity, but Not Growth

To further study the biological function of *MoPtep1*, the gene knockout mutants of *MoPtep1* were generated by a homologous recombination method and identified by PCR (Appendix A). One of the *Δmoptep1* mutant strains was randomly selected for further study. The *Δmoptep1-37* mutants showed no significant phenotypic changes compared with the wild-type in growth rate, conidiation, or colony color (Figure 4A,B), while pathogenicity analysis showed a significant reduction in the virulence of the *Δmoptep1-37* after inoculation to a three-week-old rice seedling (Figure 4C,D). To further confirm this result, the *MoPtep1* complementary strain *Δmoptep1-C91* was obtained by reintroducing the *MoPtep1* coding sequence and neomycin screening genes promoted by its native promoter into the *Δmoptep1* mutant. The results showed that complemented transformants recovered the defective virulence of *Δmoptep1* mutants (Figure 4A–D). These results indicated that *MoPtep1* is not indispensable for vegetative growth but is required for virulence in *M. oryzae*.

### 2.5. Sensitivity Analysis of Vegetative Growth of Δmoptep1 Mutant Strains under Multiple Stresses

Numerous studies show that pathogen effectors are involved in various stress responses and cell wall biosynthesis [18]. To elucidate the sensitivity of *MoPtep1* in *M. oryzae* to various stress treatments, the wild-type, the *Δmoptep1* mutant strain, and the complemented strain of *MoPtep1* were incubated on CM plates containing 10 mmol/L H_2_O_2_, 0.2 mol/L MgCl_2_, 1.5 mol/L sorbitol, 0.2 g/L Congo red (CR), and 0.01% (*w*/*v*) SDS or 0.8 μmol/L CuSO_4_. As shown in Figure 5, *Δmoptep1* mutant was not sensitive to the above stress treatment compared with the wild-type and the complemented strain of *MoPtep1*. These results suggested that *MoPtep1* maybe not be involved in the regulation of oxidative, osmotic, cell-wall-damaging, or chemical treatments.

### 2.6. Subcellular Localization of MoPtep1 in Plants

Fungal pathogens inhibit plant host immunity response via the secretion of numerous effectors. The effectors are secreted into host plant cells, different organelles are targeted, and finally, they disrupt plant resistance signaling to decrease resistance. To analyze the subcellular localization of MoPtep1 in plant cells, the transient expression system of *N. benthmena* was used to verify the localization of MoPtep1. The results demonstrated that MoPtep1-NS-GFP fluorescence fusion protein showed a bright spot signal in the *N. benthname* leaves’ cells. Further analysis showed that the fluorescence signal of MoPtep1-NS-GFP fusion protein was co-localized in peroxisomes with peroxisome marker protein DsRed-PTS1 (Figure 6). These results indicated that MoPtep1 effector protein was secreted into plant cells and may play a biological function in peroxisomes to regulate the immune response in plant cells.

### 2.7. MoPtep1 Suppressed Programmed Cell Death in N. benthamiana

Fungal effectors were secreted into host cells to inhibit or induce host cell death, increasing the pathogen virulence or suppression of host resistance [19]. To identify whether MoPtep1 affects cell death in nonhost plant *N. benthamiana*, transient expression analysis was performed through an agrobacterium-mediated method. INF1 protein from oomycete was used as the positive control. The typical cell death symptom was observed in *N. benthamiana* after INF1 protein infiltration, while no cell death was observed when infected with *A. tumefaciens* EHA105 cells expressing MoPtep1-FL-GFP or MoPtep1-NS-GFP (Figure 7A). These results suggested that MoPtep1 does not induce cell death in nonhost plant *N. benthamiana* leaves. On the contrary, the co-expression of MoPtep1-FL-GFP or MoPtep1-NS-GFP and INF1 in *N. benthamiana* resulted in the inhibition of cell death (Figure 7B). Additionally, after trypan blue staining, the infiltrated area carrying INF1 exhibited dark blue coloring, but the areas with co-filtration treatments of MoPtep1-FL-GFP or MoPtep1-NS-GFP and INF1 did not show dark blue coloring (Figure 7A,B). These results demonstrated that MoPtep1 might play important role in regulating programmed cell death in plant cells.

### 2.8. Screening and Identification of MoPtep1-Interacting Proteins

To explore how does MoPtep1 suppresses defense responses in plants, we used MoPtep1 as bait against a rice cDNA library by the yeast two-hybrid (Y2H) system. Several putative MoPtep1-interacting proteins, including momilactone A synthase-like proteins (OsHZ-7), thaumatin-like protein (OsIP-4), and plastocyanin (OsHZ-4), were identified (Appendix A). In yeast, the interaction between MoPtep1 and the candidate interacting proteins was validated (Figure 8). These results confirmed that MoPtep1 as an effector regulates immunity in rice depending on the above rice proteins through a different pathway, and the interacting mechanism needs to be further studied.

## 3. Discussion

In this study, we identified that MoPtep1 possessed a signal peptide and a cupredoxin domain, and its expression was specifically induced in the early infecting stage. Deletion of the mutant of *MoPtep1* did not impair the growth and colonial morphology but relieved the pathogenicity in the rice leaves, indicating that *MoPtep1* was required for the virulence of *M. oryzae*. The inhibition of host plant innate immunity has been regarded as the main function of bacterial, oomycete, fungal, and nematode effectors [20]. Our results suggested that *MoPtep1* plays an important role in the pathogenicity of *M. oryzae* and pathogen–host interactions. Reports showed that effectors can be the crucial factors playing important roles in pathogenicity but not in the growth phenotype. For example, the extracellular superoxide dismutase VdSOD5 is required for virulence in *Verticillium dahliae* but did not affect normal vegetative growth or colonial morphology [21]. Li reported that RxLR207 could induce ROS-mediated cell death in *N. benthamiana* and is required for the virulence of *P. capsici*, but the growth rate or mycelial morphological characteristics of the knockout mutant did not show any changes compared with the wild-type [22]. Our results also proved that effectors might play crucial roles as pathogenic factors but not in vegetative growth.

Peroxisomes are multifunctional eukaryotic organelles that play fatal roles in cellular morphology and metabolism and have been proven to be important immune response signaling organelles in human cells [23]. In recent years, numerous studies have shown that plant peroxisomes participate in numerous biological processes—for instance, stress tolerance, biomass production, plant metabolism, and pathogen defense. At present, increasing research has shown that peroxisomes played a central role in plant immunity response by regulating the defense signals and being the target organelle of pathogen effectors. For example, the potato cyst nematode *Globodera pallida* secreted effectors to maintain biotrophic interactions with its host by targeting the host peroxisomes to facilitate the infection [24]. Peroxisome protein, GOX, and the homologous proteins in *Arabidopsis*, tobacco, and rice with H_2_O_2_-producing capabilities are also proven to play important roles in plant–pathogen interaction [25]. The P8 protein of rice dwarf phytoreovirus (RDV) facilitates its infection by interacting with the GOX protein [26]. The γb protein of barley stripe mosaic virus (BSMV) also interacted with GOX protein and reduced the ROS generation in peroxisomes to promote the infection of the virus [27]. All these studies indicate that the manipulation of the plant peroxisomes’ processes is a virulence strategy shared by pathogens from multiple kingdoms of life. In our study, MoPtep1 targets the peroxisomes and could regulate the plant immunity system, as well as that of its interacting proteins, OsHZ-7, OsIP-4, and OsHZ-4. OsHZ-7 is a momilactone A synthase-like protein. It is an antibacterial compound found in husks of rice and a natural defense substance that has the activity of suppressing the growth of pathogens such as blast fungus [28,29]. OsIP-4 is a thaumatin-like protein (TLP), which plays an important role in combating plant pathogen infection. The TaTLP1-OE lines’ resistance to leaf rust and common root rot was improved [30]. In the future, we will focus on the study of the interaction between MoPtep1 and its candidate targets in host rice cells and then elucidate the mechanism that marks peroxisomes as the target organelle by effector proteins to hijack the plant immunity response.

Programmed cell death (PCD) is one of the defense-related hypersensitive responses (HR) in plant cells, and the inhibition of PCD is usually regarded as one criterion for regulating plant immunity [31]. Previous studies have shown that the potato NB-LRR receptor Gpa2 specifically inhibited the infection of *G. pallida* by targeting the GpRBP1 effector and inducing cell death. The syncytium from vascular bundle cells that provides the nutrients for nematode can be separated from the dead cells [32]. Therefore, phytopathogens must inhibit cell-death-dependent host immunity resistance for its successful inoculation. For example, *G. pallida* effector RHA1B as an E3 ubiquitin ligase suppresses HR cell death mediated by a broad range of NB-LRR receptors [33]. However, the astonishing thing was that one famous effector protein and its homologs possessed the exact opposite biology function. Transient expression of CoNIS1 and ChNIS1, but not MoNIS1, caused obvious programmed cell death, while they could significantly inhibit INF1-induced cell death and ROS production induced by PAMP in *N. benthamiana* [34]. Redundant ROS accumulation is known as one of the factors leading to cell death in the early stage of the infection site to inhibit the uptake of nutrients and to help the growth and development of the phytopathogens [35]. In our study, MoPtep1 was secreted into plant cells to suppress cell death by decreasing ROS production, depending on the peroxisomes for the *M. oryzae* survival, but the details of this mechanism need to be further studied. The present study here provides basis results to further investigate the function of MoPtep1 associated with host plant immunity.

## 4. Materials and Methods

### 4.1. Fungal Strains and Plant Growth Conditions

The wild-type *M. oryzae* strain Guy11 was used to produce the *MoPtep1* knockout (ko) mutant strains in this study. All the *M. oryzae* strains were cultured on oatmeal medium (OA) or complete medium (CM) agar plates at 25 °C conditions. The strains were grown in dark conditions for 3–4 days and transferred to light conditions for sporulation for 10–15 days. Rice plants for *M. oryzae* inoculation were grown in the growth chamber (28/26 °C, 10/14 h day/night, 85% humidity). The *Nicotiana benthamiana* plants used for cell death and subcellular localization were grown in the plant growth room (23/22 °C, 16/8 h day/night).

### 4.2. Bioinformatic and Phylogenetic Analysis of MoPtep1 Homologous Proteins

The MoPtep1 amino acid sequence was downloaded from the Ensemble Fungi database (http://fungi.ensembl.org/Magnaporthe_oryzae/Info/Index, accessed on 6 December 2021). The homologous proteins of MoPtep1 from the genomes of other organisms, including plants, oomycetes, bacteria, and pathogenic fungus, were blast and downloaded from the NCBI database (https://www.ncbi.nlm.nih.gov/). The phylogenetic tree was produced using MEGA7.0 software with the neighbor-joining method, and the multiple amino acid sequence alignment was constructed using the online software MultAlin (http://multalin.toulouse.inra.fr/multalin/multalin.html, accessed on 28 March 2000).

### 4.3. Secretion Characterization of MoPtep1

The signal peptide of MoPtep1 was predicted using SignalP-5.0 [36]. To confirm the secretory function of the supposed N-terminal signal peptide of MoPtep1, the yeast secretion system was used to validate MoPtep1 signal peptides [37]. Briefly, the full length of the *MoPtep1* and the truncated sequence that the predicted signal peptide of MoPtep1 had been deleted were amplified using the primer (Appendix A), then these two fragments were fused into the pSUC2 vector and named pSUC2-MoPtep1-FL and pSUC2-MoPtep1-NS, respectively. Then, constructs were transformed in the yeast strain YTK12. The recombinant strains were plated on CMD-W medium (0.67% yeast nitrogen base without amino acids, 0.074% -Trp DO supplement, 2% sucrose, 0.1% glucose, 1.5% agar) and YPRAA medium (1% yeast extract, 2% tryptone, 2% raffinose, and 1.5% agar with 2 μg/mL antimycin A) to detect invertase secretion. The pSUC2-Avr1b was used as the positive control, and the pSUC2-mg87 and the pSUC2-empty vector were used as the negative control. Furthermore, the secretion characterization of MoPtep1 was confirmed by the reduction of 2,3,5-triphenyltetrazolium chloride (TTC, T8170, Solarbio) to detect the insoluble red triphenylformazan, as described by Oh et al. [38]. Briefly, 5 mL of CMD-W liquid medium was inoculated with the above yeast strains that contained the specific constructs and incubated for 16 h at 30 °C. The cells were collected and resuspended with the colorless 0.1% TTC, and then the cells were transferred to a new centrifuge tube and incubated at 37 °C to observe the color change and detect the signal peptide secretion function.

### 4.4. Subcellular Localization and Cell Death Inhibition Assay

The open reading frame of *MoPtep1* (without the signal peptide and stop codon) was amplified by PCR and fused to the C-terminus of the GFP tag construct. The DsRed-PTS1 fusion construct was used as the peroxisome marker [39]. The recombinant constructs were transformed into *Agrobacterium* strain EHA105 through electroporation and then infiltrated into the *N. benthamiana* leaves. The cell death phenotype was recorded by the digital camera (Canon EOS 90D, Beijing, China), and the subcellular signal was observed in the infiltrated *N. benthamiana* leaves using the confocal microscope (Zeiss LSM Confocal, Oberkochen, Germany) at 36–48 h after infiltration. To further confirm the cell death phenotype, trypan blue staining was performed as previously described [40]. Briefly, the infiltrated *N. benthamiana* leaves were incubated in the trypan blue staining solution (25% phenol, 25% lactic acid, 25% glycerol, 24.975% water, and 0.025% trypan blue) after boiling for 20 min and left to stand at room temperature overnight; then, trichloroacetaldehyde hydrate solution was used to decolorize until the leaves were transparent.

### 4.5. Construction of MoPtep1 ko Mutant and Complemented Strains

To observe the biological function of the *Δmoptep1* mutants that were made from the wild-type strain Guy11 via the homologous recombination strategy, the split-marker strategy was performed to produce the *Δmoptep1* mutant described, and then the *M. oryzae* protoplast isolation and the transformation were used as in a previous study [40]. The candidate ko mutant strains were screened from the screening plates that included 200 μg/mL hygromycin B (400052-20ml, EMD Millipore Corp., Massachusetts, MA, USA); candidate complemented strains were screened from a 330 μg/mL G418 (G6021-5g, Macklin, Shanghai, China) plate, and the positive mutant strains were further confirmed via PCR method. All the primers used in this part are listed in Appendix A.

### 4.6. Pathogenicity Analysis

To check the phenotype of specific *M. oryzae* strains, the fungal cultures were grown in the dark for 10 days at 25 °C, and then the colony diameter statistics were taken and the cultures photographed. For pathogenicity, the wild-type and the *Δmoptep1* mutants were cultured for sporulation. The conidia were resuspended, with 1 × 10^5^ spores mL^−1^ in 0.05% Tween-20 solution. Three-week-old rice seedlings were inoculated using the above spore suspension. Inoculated plants were kept in the growth chamber with 90% humidity at 26 °C under a 12/12 h light/dark cycle. The lesions number of each treatment was recorded and photographed at 7–10 d post inoculation.

### 4.7. MoPtep1 Gene Expression Analysis

The conidia and vegetative mycelium of *M. oryzae* and the epidermis samples from barley leaves after *M. oryzae* inoculation were collected. Total RNA was extracted with TRIzol reagent (318307, Invitrogen, Carlsbad, CA, USA), and the cDNA was synthesized according to a previous study [41]. The transcriptional expression level of the *MoPtep1* was detected via ABI Prism 7500 Real-Time Detection System (Applied Biosystems, CA, USA). All data were normalized by the internal reference *actin* gene expression, and the relative abundance of transcripts was calculated by the 2^−ΔΔCt^ method. Gene expression levels were repeated for at least three technical replications. Gene-specific primers for qRT-PCR (quantitative RT-PCR) are listed in Appendix A.

### 4.8. Response of the Δmoptep1 Mutants under Different Stress Conditions

To determine the sensitivity of *Δmoptep1* mutants to stresses, mycelial growth of the wild-type strain Guy11 and *Δmoptep1* mutants were investigated after incubation on complete medium (CM) plates without or with oxidative stress agents (10 mmol/L H_2_O_2_, 0.2 mol/L MgCl_2_) [42], osmotic stress agent (1.5 mol/L sorbitol), cell wall damaging agents (0.2 g/L Congo red and 0.01% (*w*/*v*) SDS), and chemical stress agent (0.8 μmol/L CuSO_4_), as described in previous reports [43,44]. Diameters of the colonies were measured and photographed 10 days post inoculation. The experiments were repeated more than three times.

### 4.9. Yeast Two-Hybrid Assay

*MoPtep1-NS* was cloned into the vector pGBKT7 to generate BD-MoPtep1-NS. MoPtep1-NS acted as bait for prey rice proteins through a yeast two-hybrid (Y2H) system (Clontech Laboratories, Mountain View, CA, USA) derived from the yeast cDNA library. The putative MoPtep1-interacting protein genes were identified by sequencing. *OsHZ-7, OsIP-4, and OsHZ-4* were amplified by PCR using rice cDNA and then cloned into the prey vector pGADT7. Various combinations of the bait and prey constructs were cotransformed into Y2H Golden competent cells using a PEG/LiAc-based method. The selection of transformants were cultured on SD medium lacking Trp and Leu (SD-LW). Transformants were subsequently transferred to a medium lacking Trp, Leu, His, and Ade (SD-AHLW) for incubation at 28 °C for 3 days.

### 4.10. Statistical Assays

Significance assays were performed by Student’s *t*-test. All the statistic calculations were conducted by the GraphPad Prism 7 Software and SPSS25; error bars represent the standard error of more than three repeat independent experiments.

## 5. Conclusions

In this study, we identified *M. oryzae* effector protein MoPtep1, which possessed a signal peptide and a cupredoxin domain, and it was indued expression in the early infection stage. Further study showed that *MoPtep1* is required for pathogenicity, but is not necessary for the vegetative growth of *M. oryzae* and suppresses the cell death induced by INF1 in *N. benthamiana* leaves. Lastly, the Y2H system proved that MoPtep1 may be an effector to regulate plant innate immunity depending on the interacting rice proteins.

## Figures and Tables

**Figure 1 ijms-23-02515-f001:**
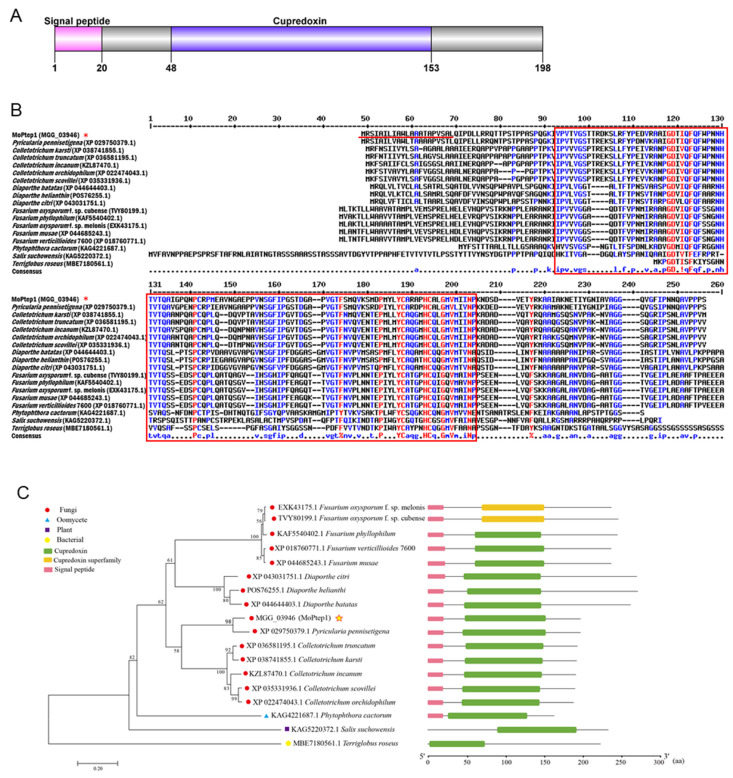
Sequence alignment and phylogenetic analysis of MoPtep1. (**A**) Structural analysis of MoPtep1. (**B**) Sequence alignment of MoPtep1 and its homologs. Potential signal peptides (red-underlined) were predicted by Signal-5.0. Residue numbers are denoted above the sequences. Cupredoxin domain is indicated by the red rectangle. (**C**) Phylogenetic tree of MoPtep1 and homologous amino acid sequences from different species. The phylogenetic tree was constructed using the neighbor-joining method. * represents the MoPtep1, The pink box represents the signal peptide, and the green and yellow boxes represent the cupredoxin domain.

**Figure 2 ijms-23-02515-f002:**
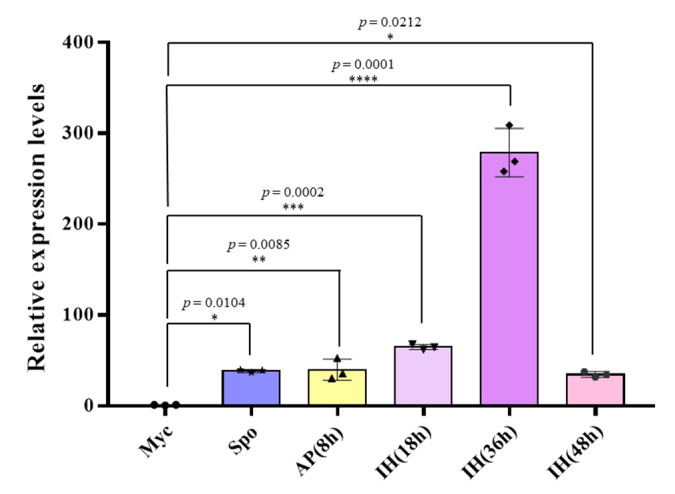
Expression pattern of *MoPtep1* in different stages. Samples from vegetative (mycelium and spore) and invasive stages (8, 18, 36, 48 hpi) of barley epidermis were performed by qRT-PCR. Myc: mycelium; Spo: spore; AP: appressoria; IH: invasive hyphae. The bar represents the standard error of the three technical duplications. Significance analysis was analyzed by Student’s *t*-test, and * indicates *p* < 0.05, ** indicates *p* < 0.01, *** indicates *p* < 0.001, and **** indicates *p* < 0.0001.

**Figure 3 ijms-23-02515-f003:**
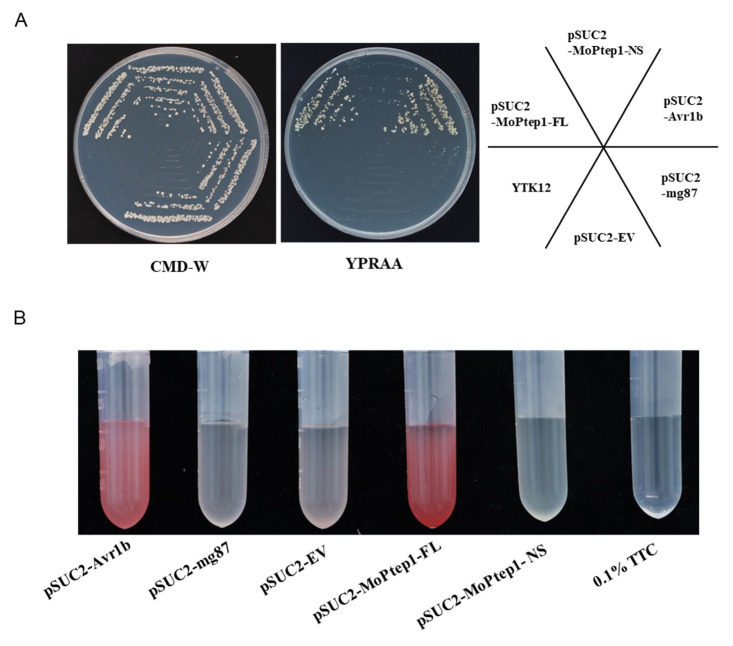
Secretion of MoPtep1 signal peptide. (**A**) The secretion of MoPtep1 signal peptide was detected by yeast signal trap assay. (**B**) TTC assay of MoPtep1 signal peptide. The recombinants pSUC2-MoPtep1-FL (full-length sequence) and pSUC2-MoPtep1-NS (no signal peptide sequence) were transformed into yeast strain YTK12. pSUC2-Avr1b was positive control, pSUC2-mg87 and pSUC2-empty vector (EV) as a negative control. All experiments were repeated three times.

**Figure 4 ijms-23-02515-f004:**
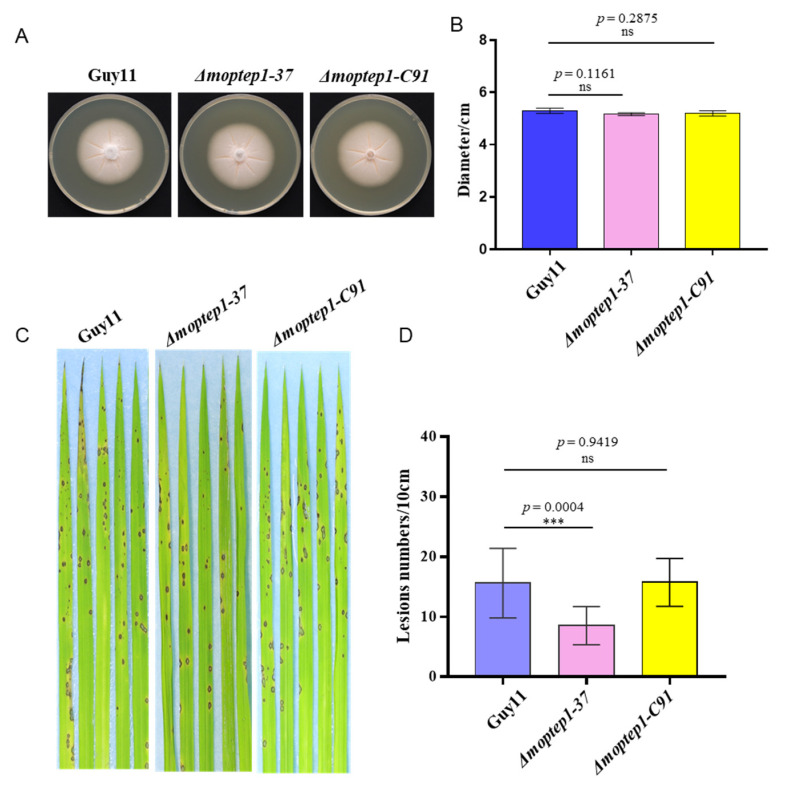
Pathogenic analysis of *Δmoptep1* mutants. (**A**) Colony morphology of wild-type strain Guy11, *Δmoptep1-37,* and *Δmoptep1-C91*. (**B**) Statistical analysis of the colony diameters. Bar indicates the standard deviation of three replicates. Significance analysis was analyzed by Student’s *t*-test, and ns means no significance. (**C**) Pathogenic results of *Δmoptep1* mutants. Conidial suspensions of Guy11, *Δmoptep1-37* mutant, and the complement strain *Δmoptep1-C91* were sprayed on rice seedlings. Diseased symptoms on leaves were photographed after 7 days post inoculation (dpi). (**D**) Statistics analysis of the number of lesions. Experiments were repeated three times. Bar indicates the standard error of 15 leaves. Significance analysis was analyzed by Student’s *t*-test; *** indicates *p* < 0.001.

**Figure 5 ijms-23-02515-f005:**
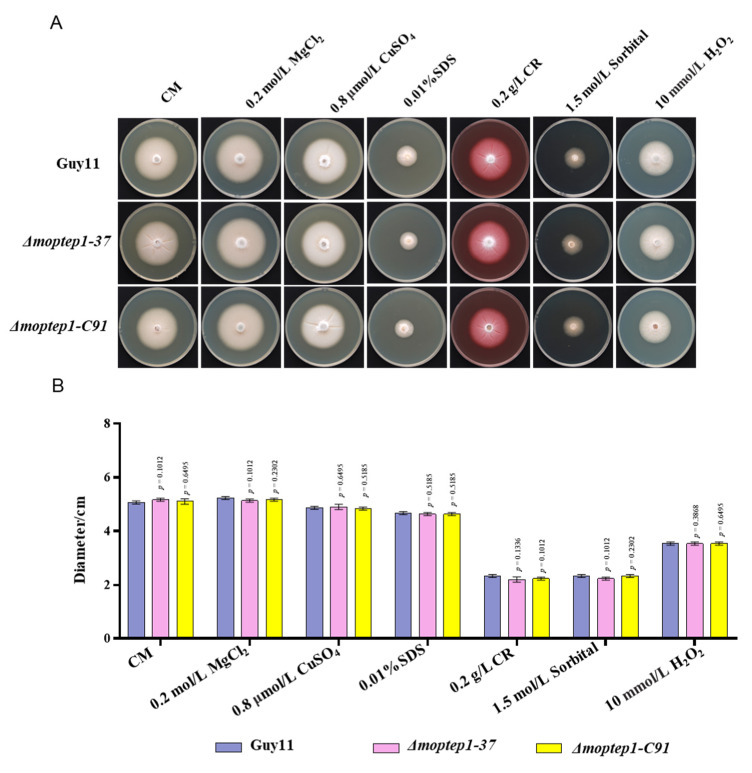
Stress analysis of *Δmoptep1* mutant. (**A**) MoPtep1 growth under multiple stresses. Guy11, *Δmoptep1-37*, and *Δmoptep1*-C91 were cultured on CM medium amended without or with stress agent (metal cations: MgCl_2_, CuSO_4_; cell wall integrity stressors: Congo red (CR) and SDS; osmotic stress: sorbitol; oxidative stress: H_2_O_2_). (**B**) Statistical analysis of the colony diameters of each strain after being cultured at 25 °C for 10 days. This experiment was repeated at least three times.

**Figure 6 ijms-23-02515-f006:**
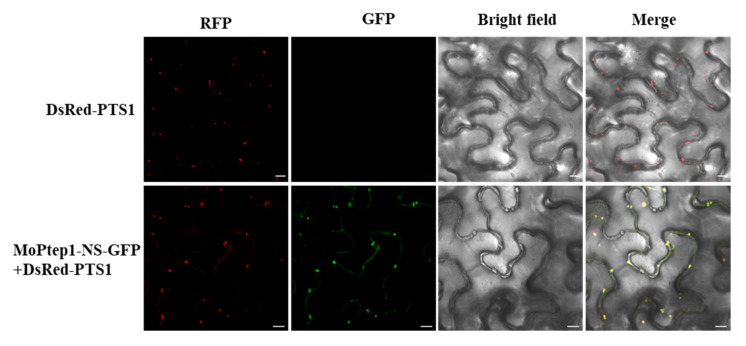
Subcellular localization of MoPtep1 in the peroxisomes in *N. benthamiana* leaf cells. Confocal images were taken from *N. benthamiana* leaf cells infiltrated by EHA105 containing the corresponding constructs at 36–48 h post inoculation (hpi). DsRed-PTS1 fusion construct was the peroxisome marker. Scale bars = 10 μm.

**Figure 7 ijms-23-02515-f007:**
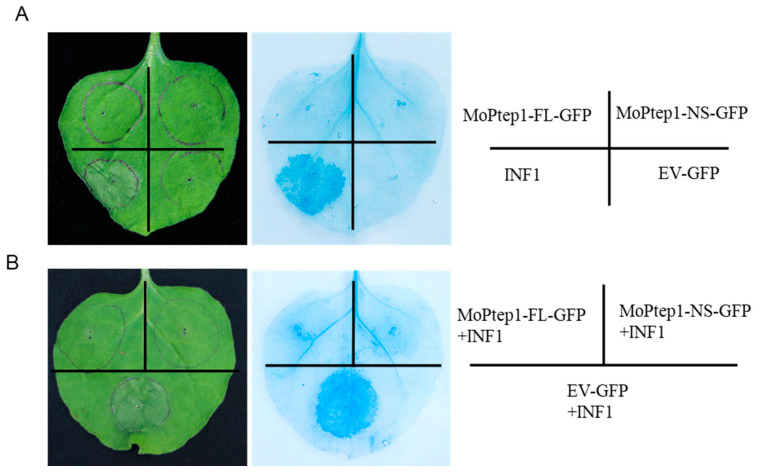
MoPtep1 inhibits Cell Death in *N. benthamiana*. (**A**) MoPtep1-FL-GFP, MoPtep1-NS-GFP, EV-GFP, and INF1 were transiently expressed in *N. benthamiana*, separately. (**B**) MoPtep1-FL-GFP and MoPtep1-NS-GFP inhibit INF1-induced cell death in *N. benthamiana*. INF1 was infiltrated into *N. benthamiana* 24 h after the MoPtep1 injection. Photographs were taken 3 days post agroinfiltration (dpa). The right pictures indicate the trypan blue staining. The agroinfiltration experiments were repeated at least three times.

**Figure 8 ijms-23-02515-f008:**
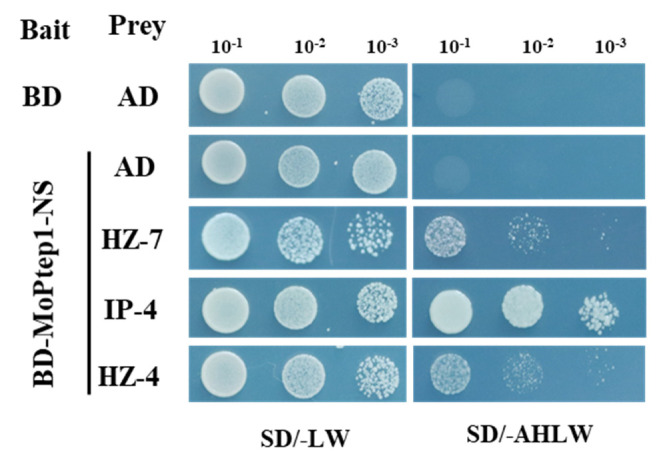
Verification of protein interactions in yeast. Analysis of protein–protein interactions between BD-MoPtep1-NS and AD-OsHZ-7, AD-OsIP-4, AD-OsHZ-4 via yeast-two-hybrid (Y2H) assay. Yeast transformants were diluted 10^−1^, 10^−2^, 10^−3^ and spotted onto selective media SD/-Trp/-Leu (SD-LW) and SD/-Trp/-Leu/-His/-Ade (SD-AHLW). Photographs were taken after incubation at 28 °C for 3 days.

## Data Availability

The data presented in this study are available on request from the corresponding authors.

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
