# Peer review of "Plant Peroxisome-Targeting Effector MoPtep1 Is Required for the Virulence of Magnaporthe oryzae"

_ijms, 2022, doi:10.3390/ijms23052515_

Round 1
Reviewer 1 Report
The work presented by Ning et al. It shows the identification and characterization of a putative secreted protein that the authors named Ptep1, conserved in fungi. Loss of Ptep1 function compromises the virulence of M. oryzae. The protein localizes to host peroxisomes and trypan blue staining experiments show that loss of the secretion of this protein induces cell death upon specific conditions.
The data are clear, and the idea of the work is easy to follow. However, my main concern is the definition of this protein as effector. To keep the article in its current form with this very basic level of experimentation, the authors cannot name this protein as an effector. I cannot accept this term with the current version of the article. I suggest or rephrase the article in a more realistic way just to focus on the identification of a protein linked to M. oryzae virulence, or to perform some additional experiments to better establish the role of Ptep1 (which I recommend to the authors).
According to the current state of the art, an effector protein is characterized by being: i) a secreted protein; ii) important for virulence and iii) interact with plant proteins (Uhse & Djamei, PLOS pathogens 2018). Also, in general, these proteins are conserved in one group of fungi but are not present in other types of fungi and plants. Assuming the role of Ptep1 in virulence, it has not been clearly shown to be secreted in M. oryzae. Furthermore, the interaction with plant proteins is a crucial experiment to define Ptep1 as an effector.
I have some suggestions for the authors:
Major changes, apart to a proper validation of the three requisites of a protein as an effector said above, are:
- It is necessary to characterize in more detail the virulence-associated phenotypes. For example, the stage of formation and penetration of the appressoria in Ptep1 mutants.
- Trypan blue or DAB staining (this one could be also done), it should be a complement to qPCR experiments analyzing the expression of plant defense genes. This information would be more precise about how the plant reacts to the loss of this potential effector.
Other suggestions:
- No proper evolutionary study has been performed, just better say that the sequence protein of Ptep1 is conserved/found in other fungi.
- The effectors in this field are always proteins, is not necessary to say: “effector protein” just say “effector”
- On page 2, “how dose the peroxisomes participate…”
- On page 3, “MoPtep1 might be an important pathogenic factor and plays crucial virulence in rice-M. oryzae interaction…” it wouldn’t say crucial
- In Figure 1, A) show percentages of identity coverage; B) please add the name of the species in each sequence; C) add an outgroup for the phylogenetic tree
- In Figure 2 and others, please substitute a, b, c, by the exact p-value and how the analysis was done.
- Epigraph 2.4, “The Δmoptep1-37 mutants sh owed no significant”
- Figure 4C, add genotype inoculated in each picture please. In my opinion, more plants inoculated are necessary to establish the role of the protein in infection since the defects are very weak.
- Study the localization of Ptep1-GFP In M. oryzae, to see if it locates at fungal peroxisomes. Also, it is crucial to check by western blot that the protein is secreted from its expression in M. oryzae.
- I don’t understand the meaning of Figure 7 as last figure, no relevant information is providing to the story.
Author Response
Comments and Suggestions for Authors
The work presented by Ning et al. It shows the identification and characterization of a putative secreted protein that the authors named Ptep1, conserved in fungi. Loss of Ptep1 function compromises the virulence of M. oryzae. The protein localizes to host peroxisomes and trypan blue staining experiments show that loss of the secretion of this protein induces cell death upon specific conditions.
Response: Thank you very much for the positive evaluations of our work. We are deeply thankful for your comments and suggestions that greatly improve our manuscript. All the concerns have been addressed in the following sections.
The data are clear, and the idea of the work is easy to follow. However, my main concern is the definition of this protein as effector. To keep the article in its current form with this very basic level of experimentation, the authors cannot name this protein as an effector. I cannot accept this term with the current version of the article. I suggest or rephrase the article in a more realistic way just to focus on the identification of a protein linked to M. oryzae virulence, or to perform some additional experiments to better establish the role of Ptep1 (which I recommend to the authors).
Response: Thank you very much for your suggestions, and we have responded to your comments point by point showed as follows, and added experiment results in our manuscript to prove that Ptep1 is one of the effector proteins.
According to the current state of the art, an effector protein is characterized by being: i) a secreted protein; ii) important for virulence and iii) interact with plant proteins (Uhse & Djamei, PLOS pathogens 2018). Also, in general, these proteins are conserved in one group of fungi but are not present in other types of fungi and plants. Assuming the role of Ptep1 in virulence, it has not been clearly shown to be secreted in M. oryzae. Furthermore, the interaction with plant proteins is a crucial experiment to define Ptep1 as an effector.
Response: Thank you very much for your suggestions. As your suggestions, we have added the result that we screened the rice cDNA library using Ptep1 as the bait, and we have got three Ptep1 candidate interacting proteins in “2.8 Screening and identification of MoPtep1 interacting proteins”, and this result could prove that Ptep1 is one of the effector proteins.
I have some suggestions for the authors:
Major changes, apart to a proper validation of the three requisites of a protein as an effector said above, are:
It is necessary to characterize in more detail the virulence-associated phenotypes. For example, the stage of formation and penetration of the appressoria in Ptep1 mutants.
Response: Thanks for your suggestion. We have identified the different inoculation stages (including formation and penetration of the appressoria) of the Ptep1 mutants using different methods, but we did not find the obvious difference between the Ptep1 mutants and the wild type. So, Ptep1 maybe control the virulence of M.oryzae through more complex mechanisms.
Trypan blue or DAB staining (this one could be also done), it should be a complement to qPCR experiments analyzing the expression of plant defense genes. This information would be more precise about how the plant reacts to the loss of this potential effector.
Response: Thanks for your suggestion. All the experiments that you said, including the staining and the qPCR analysis we have performed in the process of studying the function of Ptep1, but it is a pity, we did not get very consistent results. So, to study the specific functions, we screeded the Ptep1 interacting protein in rice, and three rice proteins have been proved to interact with Ptep1 in the yeast system, all our results can prove that Ptep1 is one of the effector proteins.
Other suggestions:
No proper evolutionary study has been performed, just better say that the sequence protein of Ptep1 is conserved/found in other fungi.
Response: According to your suggestion, we revised “MoPtep1 was evolutionarily conserved in plant pathogenic fungi” into “the sequence protein of MoPtep1 is conserved in pathogenic fungi.” at line 125-126.
The effectors in this field are always proteins, is not necessary to say: “effector protein” just say “effector”
Response: Thanks for your suggestion, we changed the "effector protein" into "effector".
On page 2, “how dose the peroxisomes participate…”
Response: Thank you very much. We revised “how dose the peroxisomes participate…” into “how does the peroxisomes participate” at line 85
On page 3, “MoPtep1 might be an important pathogenic factor and plays crucial virulence in rice-M. oryzae interaction…” it wouldn’t say crucial
Response: According to your suggestion, we have changed the sentence “MoPtep1 might be an important pathogenic factor and plays crucial virulence in rice-M. oryzae interaction…” to “MoPtep1 might be an important pathogenic factor and plays a role virulence in rice-M. oryzae interaction…” at line 101-102.
In Figure 1, A) show percentages of identity coverage; B) please add the name of the species in each sequence; C) add an outgroup for the phylogenetic tree
Response: Thanks for your suggestion. According to your suggestion, we add the percentage of identity in the revised manuscript at line 111- 112., B: added the name of each sequence in Figure 1.B. C: No homologous sequences have been found in other pathogenic genera or species. The sequences of plants and bacteria can be used as outgroups to determine the roots of the evolutionary tree.
In Figure 2 and others, please substitute a, b, c, by the exact p-value and how the analysis was done.
Response: Thanks for your suggestion. According to your suggestion, we modified the significance analysis part in Figure 2 and Figure 4 by changing a, b and c to P-values. Some modifications have also been made to the figure annotations.
Epigraph 2.4, “The Δmoptep1-37 mutants sh owed no significant”
Response: Thanks for your suggestion, we change the above sentence into “The Δmoptep1-37 mutants showed no significant”
Figure 4C, add genotype inoculated in each picture please. In my opinion, more plants inoculated are necessary to establish the role of the protein in infection since the defects are very weak.
Response: Thanks for your suggestion, We inoculated more mutant genotypes in the early stage of the experiment. However, due to time reasons, we could not add more inoculated phenotypes of the complementary strain.
Pathogenic analysis of Δmoptep1 mutants.
(A) Pathogenic results of Δmoptep1 mutants. Conidial suspensions of Guy11 and Δmoptep1 mutants were sprayed on rice seedlings. Diseased symptoms on leaves were photographed after 7 days post inoculation (dpi). (B) Statistics analysis of the number of lesions. Experiments were repeated three times. Bar indicates the standard error of 15 leaves. Small letter indicates a significant difference (P < 0.05).
Study the localization of Ptep1-GFP In M. oryzae, to see if it locates at fungal peroxisomes. Also, it is crucial to check by western blot that the protein is secreted from its expression in M. oryzae.
Response: Thanks for your suggestion. According to your suggestion, We detected the secretion of the MoPtep1 in M. oryzae, and the results are shown as follows: the Ptep1-GFP fusion protein about 48 kDa could be identified, at the same time the free GFP protein signal also been identified in the mycelium. Meanwhile, only the free GFP protein signal could be identified in the culture medium supernatant. This result suggested that MoPtep1 was secreted, but was degraded by some unknown reasons.
Detection of MoPtep1 secretions in M. oryzae. Guy11, Ptep1-GFP fusion overexpressing strains were cultured in CM liquid medium at 28℃ and 210 rpm for 48 h. Mycelia and culture medium supernatant were collected, and proteins were extracted for western blot detection. Myc: Mycelia, Sup: supernatant.
I don’t understand the meaning of Figure 7 as the last figure, no relevant information is provided to the story.
Response: Thanks for your suggestion, we have moved Figure 7 to Figure 5 in the revised manuscript. The sensitivity of MoPtep1 ko mutant strain to a series of stresses was analyzed.

Reviewer 2 Report
The manuscript is well written and presented. I only have a few minor comments.
Minor corrections
Page 3
(thereafter named MoPtep1,MoPtep1, Peroxisomes-targeted effector protein 1).
Pls correct, there after, and turns the underline feature off
…. f. sp. Melonis, Pls correct it, f. sp. melonis
…. f. sp. Cubense, Pls correct it, f. sp. cubense
Page 7
Figure 4. Use the s.e.d. not the standard deviation. Pls check if Guy11 is different from Δmoptep1-37 mutant
Page 10
Figure 7. please present bars in all columns
Page 11
Verticillium dahlia, pls correct it, Verticillium dahliae
P. capsica, pls correct it, P. capsici
Arabidopsis not in italics
Author Response
Comments and Suggestions for Authors
The manuscript is well written and presented. I only have a few minor comments.
Response: Thank you very much for the positive evaluations of our work. We are deeply thankful for your comments and suggestions. All the concerns have been addressed in the following sections.
Minor corrections
Page 3
(thereafter named MoPtep1,MoPtep1, Peroxisomes-targeted effector protein 1).
Pls correct, there after, and turns the underline feature off
Response: According to your suggestion, we got rid of the underline at line 106.
…. f. sp. Melonis, Pls correct it, f. sp. melonis
…. f. sp. Cubense, Pls correct it, f. sp. cubense
Response: Thanks for your suggestion. According to your suggestion, we revised “f. sp. Melonis” into “f. sp. melonis” at line 112; “f. sp. Cubense” into “f. sp. cubense” at line 113.
Page 7
Figure 4. Use the s.e.d. not the standard deviation. Pls check if Guy11 is different from Δmoptep1-37 mutant
Response: Thanks for your suggestion. According to your suggestion, we changed the "standard deviation" into "standard error". The experimental data were analyzed by software (GraphPad Prism 7 Software and SPSS25), and there were errors in the presentation process, which have been modified according to your suggestions.
Page 10
Figure 7. please present bars in all columns
Response: Thanks for your suggestion. We added bas for all columns.
Page 11
Verticillium dahlia, pls correct it, Verticillium dahliae
- capsica, pls correct it, P. capsici
Arabidopsis not in italics
Response: Thanks for your suggestion. According to your suggestion, we revised “Verticillium dahlia” into “erticillium dahliae”, “P. capsica” into “P. capsici” and Arabidopsis into italics in our manuscript.

Reviewer 3 Report
The manuscript entitled "Plant peroxisomes targeting effector protein MoPtep1 is re-quired for the virulence of Magnaporthe oryzae." has been submitted by Nung etal as an original article.
The authors identify the Magnaporthe oryzae Ptep1 (Peroxisomes-targeted effector protein 1) as a secreted protein that targets plant peroxisomes in order to inhibit cell death.
The manuscript is of interest and most of the conclusions are supported by the presented data.
1) The authors should add an important control concerning the targeting of the protein. The authors suggest that the N-terminal sequence of Ptep1 exhibits a peroxisomal targeting signal. The authors should use a truncated version of the protein that lacks this sequence in order to proof its function as peroxisomal targeting signal in co-localization assays with peroxisomes.
2) The N-terminal sequence suggests that this is a PTS2 signal. The authors should analyze whether the targeting of Ptep1 is depending on the presence of the PTS2 receptor PEX7.
3) It would be impoirtant to analyze whether the truncated version of Ptep1 is still capable to fulfill the function of the full-length protein.
Author Response
Comments and Suggestions for Authors
The manuscript entitled "Plant peroxisomes targeting effector protein MoPtep1 is re-quired for the virulence of Magnaporthe oryzae." has been submitted by Nung etal as an original article.
The authors identify the Magnaporthe oryzae Ptep1 (Peroxisomes-targeted effector protein 1) as a secreted protein that targets plant peroxisomes in order to inhibit cell death.
The manuscript is of interest and most of the conclusions are supported by the presented data.
Response: Thank you very much for the positive evaluation and valuable suggestion on our work. All the concerns have been addressed in the following sections.
1) The authors should add an important control concerning the targeting of the protein. The authors suggest that the N-terminal sequence of Ptep1 exhibits a peroxisomal targeting signal. The authors should use a truncated version of the protein that lacks this sequence in order to proof its function as peroxisomal targeting signal in co-localization assays with peroxisomes.
Response: Thanks for your suggestion. In subcellular localization experiments, we performed experiments using the sequence MoPtep1-NS which the signal peptide was removed. At the same time, MoPtep1-FL was also checked for localization observation, but no obvious peroxisome signal was observed, so the full-length subcellular localization images were not included in the manuscript.
2) The N-terminal sequence suggests that this is a PTS2 signal. The authors should analyze whether the targeting of Ptep1 is depending on the presence of the PTS2 receptor PEX7.
Response: Thanks for your suggestion. In this study, we only used PTS1 as a marker protein, so we did not pay much attention to its receptor protein. In subsequent experiments, we have screened the interacting proteins of MoPtep1 in rice, and whether it depends on PTS2 receptor PEX7 needs further analysis. Thank you very much for your advice.
3) It would be impoirtant to analyze whether the truncated version of Ptep1 is still capable to fulfill the function of the full-length protein.
Response: Thanks for your suggestion. We have studied both full-length and no-signal peptide sequences of Ptep1.

Round 2
Reviewer 1 Report
The paper has been rephrased to better fit the results shown. I have no more suggestions.
Author Response
The paper has been rephrased to better fit the results shown. I have no more suggestions.
Response: Thank you very much for the positive evaluations of our work. We are
deeply thankful for your comments and suggestions that greatly improve our
manuscript.
Reviewer 3 Report
The authors Ning et al. have submitted a revised version of the article
"Plant peroxisomes targeting effector protein MoPtep1 is re-quired for the virulence of Magnaporthe oryzae.".
The authors have addressed most of my points. However, the most important problem is still not solved.
The authors find that both the full length protein and the truncated version lacking the signal peptide co-localize with plant peroxisomes. This means that the so-called signal peptide may be only involved in secretion from the fungal cell and not in the process of the targeting to plant peroxisomes.
The authors could not rule out that the observed co-localization of full-length and truncated MoPtep1-GFP with the peroxisomal marker RFP-PTS1 indeed represents the result of a physiologic targeting of the protein.
a) The authors should define the peroxisomal marker, which they call "PTS1-RFP" (suggesting a N-terminal fusion), in more detail: Was the PTS1-sequence fused to the N-terminus or the C-terminus of the RFP ? This is important, because the PTS1 is found at the C-Terminus of peroxisomal matrix proteins.
b) The authors should add a control that disrupts the targeting of the MoPtep1-versions to peroxisomes. This could be done with the described co-localization experiment using PEX5 and/or PEX7 deficient plant cells (no known PTS receptors) or PEX19 deficient plant cells (no defined peroxisomal membranes).
Author Response
Comments and Suggestions for Authors
The authors Ning et al. have submitted a revised version of the article
"Plant peroxisomes targeting effector protein MoPtep1 is re-quired for the virulence of Magnaporthe oryzae.".
The authors have addressed most of my points. However, the most important problem is still not solved.
The authors find that both the full-length protein and the truncated version lacking the signal peptide co-localize with plant peroxisomes. This means that the so-called signal peptide may be only involved in secretion from the fungal cell and not in the process of the targeting to plant peroxisomes.
Response: Thank you for your suggestion. In our study, the localization of the MoPtep1 full-length protein and that without the signal peptide were different. We observed that the localization of the full-length protein fused with GFP was in the membrane system, but the signal of MoPtep1 without the signal peptide fused with GFP was spot-like and co-locate with the signal of DsRed-PTS1. These results suggest that the N-terminal signal peptide sequence of MoPtep1 not only has a secretory function but also affect the subcellular localization of MoPtep1.
Supplemental Fig. 1 Subcellular localization of MoPtep1-FL in the membrane system in N. benthamiana leaf cells.
Figure. 6 Subcellular localization of MoPtep1-NS in the peroxisomes in N. benthamiana leaf cells.
The authors could not rule out that the observed co-localization of full-length and truncated MoPtep1-GFP with the peroxisomal marker DsRed-PTS1 indeed represents the result of a physiologic targeting of the protein.
Response: Thank you very much for your suggestions. In previous studies, peroxisome localization of M-LP (MPV17-like protein) and ChEC89 was verified using DsRed2-PTS1 and PTS2- DsRed, a fluorescent marker of peroxisomes [1,2]. We agree with you that we cannot rule out that MoPtep1-NS-GFP and DsRed-PTS1 co-localization are indeed accurate Ptep1 localization depending on our present results. In the next step, we will verify whether MoPtep1-NS-GFP entered the peroxisome due to the co-injection of MoPtep1-NS-GFP and DsRed-PTS1.
- a) The authors should define the peroxisomal marker, which they call "PTS1-RFP" (suggesting a N-terminal fusion), in more detail: Was the PTS1-sequence fused to the N-terminus or the C-terminus of the RFP? This is important, because the PTS1 is found at the C-Terminus of peroxisomal matrix proteins.
Response: You are right that PTS1 is fused at the C-terminus of the DsRed. The DsRed-PTS1 plasmid used in the study was derived from Marine and Agricultural Biotechnology Laboratory, Institute of Oceanography, Minjiang University [3].
According to your suggestion, we have changed "PTS1-RFP" into "DsRed-PTS1" in the manuscript.
- b) The authors should add a control that disrupts the targeting of the MoPtep1-versions to peroxisomes. This could be done with the described co-localization experiment using PEX5 and/or PEX7 deficient plant cells (no known PTS receptors) or PEX19 deficient plant cells (no defined peroxisomal membranes).
Response: Thank you very much for your advice. Peroxisome targeting signals (PTS) is conserved in eukaryotes, mainly including PTS1, PTS2, and PTS3. The consistent sequence of PTS1 is (S/C/A)(K/R/H)(L/M), and it mainly depends on Pex5 to enter the peroxisome. PTS2 was initially considered as a non-peptide with the sequence of -RLX5(H/Q)L- and was later changed to -(R/K)(L/V/I)X5(H/Q)(L/A)-, which can be recognized by the Pex7 receptor. After the sequence analysis of MoPtep1-GFP, neither PTS1 C-terminal tripeptide sequence nor PTS2 motifs were found. Therefore, MoPtep1-NS-GFP may not be dependent on PTS1 receptor Pex5 or PTS2 receptor Pex7. How does MoPtep1-GFP enter the peroxisome needs to be further demonstrated by follow-up experiments.
Recent studies have shown COX4 as a mitochondrial marker to observe the subcellular localization of OsDjA9, HDEL as an endoplasmic reticulum marker to observe the subcellular localization of the effector MoCDIP4 [4–6], and PWL2 as a nuclear marker to observe MoHTR1 and MoHTR2 subcellular localization [7,8]. COX4, HDEL, and PWL2 were all used as markers only. Referring to these studies, we used DsRed-PTS1 as a marker to label the peroxisomes is feasible.
As a further systematic study, I couldn't agree more with your suggestion that adding an important control regarding protein targeting, description of colocalization experiments using PEX5 and/or PEX7, or PEX19 deficient plant cells. However, we do not have relevant materials at present, and this part study will be done in the future study. Thank you again for your suggestion.
- Iida, R.; Yasuda, T.; Tsubota, E.; Takatsuka, H.; Masuyama, M.; Matsuki, T.; Kishi, K. M-LP, Mpv17-like Protein, Has a Peroxisomal Membrane Targeting Signal Comprising a Transmembrane Domain and a Positively Charged Loop and Up-regulates Expression of the Manganese Superoxide Dismutase Gene. J. Biol. Chem. 2003, 278, 6301–6306, doi: 10.1074/jbc.M210886200.
- Robin, G.P.; Jochen, K.; Ulla, N.; Lisa, C.; Jean-Félix, D.; Nicolas, L.; O’Connell, R.J. Subcellular Localization Screening of Colletotrichum higginsianum Effector Candidates Identifies Fungal Proteins Targeted to Plant Peroxisomes, Golgi Bodies, and Microtubules. Front. Plant Sci. 2018, 9, 562, doi:10.3389/fpls.2018.00562.
- Chen, Z.; Zheng, W.; Chen, L.; Li, C.; Liang, T.; Chen, Z.; Xu, H.; Han, Y.; Kong, L.; Zhao, X.; et al. Green Fluorescent Protein- and Discosoma sp. Red Fluorescent Protein-Tagged Organelle Marker Lines for Protein Subcellular Localization in Rice. Front Plant Sci. 2019, 10, 1421, doi:10.3389/fpls.2019.01421.
- Nelson, B.K.; Cai, X.; Nebenführ, A. A multicolored set of in vivo organelle markers for co-localization studies in Arabidopsis and other plants. Plant J. 2007, 51, 1126–1136, doi:10.1111/j.1365-313X.2007.03212.x.
- Napier, R.M.; Fowke, L.C.; Hawes, C.; Lewis, M.; Pelham, H.R. Immunological evidence that plants use both HDEL and KDEL for targeting proteins to the endoplasmic reticulum. J. Cell Sci. 1992, 102, 261–271, doi:10.1242/jcs.102.2.261.
- Xu, G.; Zhong, X.; Shi, Y.; Liu, Z.; Jiang, N.; Liu, J.; Ding, B.; Li, Z.; Kang, H.; Ning, Y.; et al. A fungal effector targets a heat shock-dynamin protein complex to modulate mitochondrial dynamics and reduce plant immunity. Sci. Adv. 2020, 6, doi:10.1126/sciadv.abb7719.
- Khang, C.H.; Berruyer, R.; Giraldo, M.C.; Kankanala, P.; Park, S.-Y.; Czymmek, K.; Kang, S.; Valent, B. Translocation of Magnaporthe oryzae effectors into rice cells and their subsequent cell-to-cell movement. Plant Cell. 2010, 22, 1388–1403, doi:10.1105/tpc.109.069666.
- Kim, S.; Kim, C.Y.; Park, S.Y.; Kim, K.T.; Jeon, J.; Chung, H.; Choi, G.; Kwon, S.; Choi, J.; Jeon, J.; et al. Two nuclear effectors of the rice blast fungus modulate host immunity via transcriptional reprogramming. Nat. Commun. 2020, 11, 5845, doi:10.1038/s41467-020-19624-w.

Round 3
Reviewer 3 Report
The manuscript entitled "Plant peroxisomes targeting effector protein MoPtep1 is re-quired for the virulence of Magnaporthe oryzae." has been submitted as revised version by Ning et al.
The authors have addressed the essential points in the new version of the manuscript.
Author Response
Comments and Suggestions for Authors
The manuscript entitled "Plant peroxisomes targeting effector protein MoPtep1 is re-quired for the virulence of Magnaporthe oryzae." has been submitted as revised version by Ning et al.
The authors have addressed the essential points in the new version of the manuscript.
Response: Thank you very much for your suggestions and the positive evaluation of our modification work on the manuscript.